# How Does Nutrition Affect the Epigenetic Changes in Dairy Cows?

**DOI:** 10.3390/ani13111883

**Published:** 2023-06-05

**Authors:** Ana Lesta, Pablo Jesús Marín-García, Lola Llobat

**Affiliations:** 1MMOPS Research Group, Departamento Producción y Sanidad Animal, Salud Pública y Ciencia y Tecnología de los Alimentos, Facultad de Veterinaria, Universidad Cardenal Herrera—CEU, CEU Universities, 46115 Valencia, Spain; ana.lesta@alumnos.uchceu.es; 2Department of Animal Production and Health, Veterinary Public Health and Food Science and Technology (PASAPTA), Facultad de Veterinaria, Universidad Cardenal Herrera—CEU, CEU Universities, 46113 Valencia, Spain; pablo.maringarcia@uchceu.es

**Keywords:** cow, DNA methylation, epigenetic changes, histone deacetylation, milk production, miRNA

## Abstract

**Simple Summary:**

Nutrition plays a key role in the epigenetic regulation of gene expression in dairy cows. Epigenetic alterations refer to the changes in gene expression that are not caused by changes in the DNA sequence itself, but rather by modifications to the DNA molecule or the proteins that interact with it. These modifications can be influenced by environmental factors such as diet and have a deep impact on the health and productivity of dairy cows. This work summarizes the main causes of nutrition that produce epigenetic changes in dairy cattle.

**Abstract:**

Dairy cows require a balanced diet that provides enough nutrients to support milk production, growth, and reproduction. Inadequate nutrition can lead to metabolic disorders, impaired fertility, and reduced milk yield. Recent studies have shown that nutrition can affect epigenetic modifications in dairy cows, which can impact gene expression and affect the cows’ health and productivity. One of the most important epigenetic modifications in dairy cows is DNA methylation, which involves the addition of a methyl group to the DNA molecule. Studies have shown that the methylation status of certain genes in dairy cows can be influenced by dietary factors such as the level of methionine, lysine, choline, and folate in the diet. Other important epigenetic modifications in dairy cows are histone modification and microRNAs as regulators of gene expression. Overall, these findings suggest that nutrition can have a significant impact on the epigenetic regulation of gene expression in dairy cows. By optimizing the diet of dairy cows, it may be possible to improve their health and productivity by promoting beneficial epigenetic modifications. This paper reviews the main nutrients that can cause epigenetic changes in dairy cattle by analyzing the effect of diet on milk production and its composition.

## 1. Introduction

Cattle produced around 930 million tons of milk in 2022, up by 0.6 percent from 2021. The Food and Agriculture Organization of the United Nations expects an increase in milk production by more than 15 million tons per year by 2030, mainly in developing countries [1]. In addition, milk production also plays a relevant role in economic development and poverty mitigation [2]. In relation to human nutrition, milk and dairy products are one of the main sources of high-quality protein; vitamins A, D3, B1, B2, B6, and B12; and other micronutrients such as calcium, phosphorus, selenium, and potassium [3,4]. The milk compounds and types of fat in dairy are involved with bone health, cardiovascular disease, and other conditions, including immune development in children [4,5,6,7]. The quality and quantity of these and other relevant nutrients depend on the cow’s health and nutrition, and other factors such as milk processing.

Increasing productivity while maintaining quality through genetic selection, management of cattle, and nutritional strategies is a topic that is well studied [8,9,10]. In recent years, genetic selection in dairy cows has improved milk production and quality. Other traits such as health, fertility, embryo production, and mastitis resistance have also been improved through genetic selection in dairy cows [11,12,13]. The application of omics technologies has achieved notable improvements in these and other traits in dairy cattle [14]. New management systems of cows have had a great impact on the reproduction performance [15], resistance to infections [16], and health [17]. In relation to nutrition, different strategies were assessed to increase milk production and fertility parameters [18], and to reduce methane emissions [19], among other benefits.

Epigenetic mechanisms regulate gene expression [20]. The impact of the environment and nutrition on epigenetic changes was demonstrated in animals [21]. For example, high levels of renal renin expression were shown to regulate gene expression in animals [22], whereas in humans, a methyl-group donor could modulate gene expression via DNA and histone methylation [23]. In cows, epigenetic changes could influence not only their performance, but also their offspring. The aim of this review is to update the current knowledge about the effects of nutritional management on epigenetic mechanisms in dairy cattle.

## 2. Nutrition in Dairy Cows

Nutrient requirements in cattle are dependent on the physiological stage, breed, and environmental conditions, among other factors. Therefore, dairy cows need a diet that supplies the energy and nutrients needed for high milk production. Energy, carbohydrates, amino acids (AAs), fatty acids, minerals, vitamins, and water are all nutrients required by lactating dairy cows to meet the demand of the mammary gland to produce milk and its components [18]. The energy requirements are calculated by estimating the maintenance and production needs. A 10% safety margin is included in maintenance requirements to cover the energy costs resulting from normal activity. Production requirements should consider the chemical composition of milk, especially in relation to fat content, not only because it has a high energy value compared to other components, but also because it varies considerably between animals and/or farms [24]. The current stage of the productive cycle should be considered as well, since the cow’s dry matter intake and milk production vary considerably throughout the entire cycle [18,25,26,27]. High levels of milk production require an increase in energy in the diet. Figure 1 shows the daily net energy requirements depending on milk production and milk composition.

Lactation is the most demanding biological process. This process requires the cows to consume enough nutrients to produce milk and to gain weight. So, if the diets are deficient in any nutrient, the milk production and its components will decrease [29,30]. However, diets with excessive amounts of nutrients will decrease the efficiency of its nutrient utilization, increase nutrient excretion into the environment, increase the cost of milk production, decrease the profits for dairy producers, and increase the costs for the consumers of dairy products [29].

Bovine milk is a nutritionally rich, chemically complex biofluid consisting of hundreds of different components, primarily water and triglycerides [31]. It is the secretory product of mammary epithelial cells (MEC), where the vast majority of the compounds are synthesized from blood precursors [32]. It is essential to highlight the key role that a rich and balanced diet plays in milk production, although there are certain features concerning rumen physiology and microbiome that should be considered when it comes to making up the feed ration [33,34].

Considerable progress was made in understanding the protein and AAs nutrition of dairy cows. Worldwide, most producers and nutritionists still consider only crude protein (CP) when evaluating protein diets and animal requirements. There is a mechanism of ruminal protein degradation by rumen bacteria and protozoa. It was shown that ammonia released from AA degradation in the rumen is used for bacterial protein formation, and that urea can be a useful nitrogen (N) supplement when low-protein diets are used [34]. 

Dietary protein generally refers to CP, which is defined for feedstuffs as the nitrogen content, which is obtained by multiplying the nitrogen content by a factor of 6.25, since most proteins contain 16% N. The CP content includes both protein and nonprotein N (NPN). Feedstuffs vary widely in their relative proportions of CP and NPN, in the rate and extent of the ruminal degradation of the protein, and in the intestinal digestibility and its AA composition of ruminal undegraded feed protein. The NPN in feedstuffs and supplements such as urea and ammonium salts are degraded completely in the rumen. Nevertheless, the goals of ruminant protein nutrition are to provide adequate amounts of rumen-degradable protein (RDP) for optimal ruminal efficiency and to obtain the desired animal productivity with a minimum amount of dietary CP. In fact, an increase in milk production and composition requires an increase in crude protein in the diet (Figure 2).

Amino acid requirements have only been determined generically for methionine (Met) and lysine (Lys), although under some conditions, arginine and histidine may also limit milk production. The AA requirements for adult dairy cattle are known with little certainty, but both the NRC (2001) and the INRA (2007) recommend the contribution of 2.4 and 7.2% of the total metabolizable protein to be Met and Lys, respectively [28,35].

To comply with the AA requirements, the use of rumen-protected AAs (RPAAs) was evaluated. Met and Lys supplementation was widely studied, showing benefits related to milk and its protein yield [18]. However, some works suggest that the protein needs of both the animal and the rumen were overestimated, and that lower levels than those currently recommended would be sufficient for maintaining high production levels with higher N retention efficiencies and lower N emissions into the environment, with balanced AA levels [19,34]. Figure 3 shows the daily RDP requirements in different reproductive stages related to milk production and milk composition.

## 3. Epigenetic Regulators

### 3.1. DNA Methylation

In mammals, DNA methylation plays a crucial role in regulating different biological functions, including chromosomal stability, genomic imprinting, and X-chromosome inactivation [36]. This process involves the addition of methyl groups to the cytosine of CpG islands (Figure 4) and is catalyzed by the following three conserved enzyme families: DNA methyltransferase (DNMT) 1, DNMT3a, and DNMT3b [37,38,39]. These three enzymes play crucial roles in DNA methylation, an epigenetic modification process in which a methyl group is added to the DNA molecule. DNA methylation helps regulate gene expression and is involved in various cellular processes, including development, differentiation, and genomic stability. DNMT1 (DNA methyltransferase 1) is primarily responsible for maintaining DNA methylation patters during DNA replication. It recognizes hemimethylated DNA and adds a methyl group to the unmethylated strand, ensuring the faithful transmission of DNA methylation patterns to daughter cells. DNMT3a (DNA methyltransferase 3a) is one de novo DNA methyltransferase. It is involved in establishing new DNA methylation patterns during development and cellular differentiation. DNMT3A can methylate previously unmethylated DNA regions and is crucial for embryonic development, hematopoiesis, and neuronal differentiation. DNMT3b (DNA methyltransferase 3b) shares functional similarities with DNMT3a and is involved in establishing DNA methylation patterns during development and cellular differentiation. DNMT3b is particularly important for the methylation of repetitive sequences, such as transposable elements in the genome. These three enzymes collectively contribute to the dynamic regulation of DNA methylation patterns in cells, playing critical roles in normal development, gene expression, and disease processes [40,41,42,43,44,45].

For these enzymes to perform their function, they require a methyl donor, which can be influenced by methyl groups (choline, betaine, methyl-folate, or methionine) obtained from the diet. These substances serve as precursors to the universal methyl donor, S-adenosylmethionine (SAM) [44,46,47,48]. One example of the impact of methyl groups on enzyme function is the regulation of the insulin-like growth factor II (*IGF2*) gene. The expression of *IGF2* depends on the methylation of a specific region, *Igf2DMR2*, located in the *H19* gene. When pregnant rats are fed a choline-deficient diet, hypermethylation occurs in these regions, resulting in the inhibition of *H19* and an increase in the *IGF2* expression [49]. A prime example of this *EGF2* regulation and its relationship with nutrition occurred in the Dutch Hunger Winter of 1944/1945. A reduction in the DNA methylation in the *IGF2* gene was observed in the offspring of pregnant women who suffered from famine [50]. Several studies indicate the importance of nutrition in determining epigenetic marks in livestock. Murdoch et al. (2016) published an extensive review that explores the effects of nutrient quality and quantity on methylation in different livestock species [51]. Among the various nutrients, methyl-group donors such as choline, folate, and betaine have a significant effect on DNA methylation. For example, betaine injections in eggs were shown to increase DNA methylation, regulating hepatic cholesterol metabolism in chicks [52]. Pregnant sows fed with betaine-supplemented diets were also found to have piglets with modified methylation patterns [53]. Sheep that were fed diets deficient in vitamin B12, folate, and Met were linked to hypomethylation in their offspring, which can affect birth weight, immune responses, and blood pressure levels [50].

The most economically important traits in dairy cows are related to quantitative trait loci (QTL), including milk yield and milk components. Some of these QTL are regulated by epigenetic mechanisms, including methylation regulation, and may affect phenotypic variation in livestock production [54,55]. Therefore, it is necessary to conduct studies on the epigenetic regulation, particularly through DNA methylation, which are traits of interest in livestock production. Such studies should focus on how animal nutrition affects these methylation changes.

### 3.2. Histone Modifications

Histone post-translational modifications are chemical modifications that occur on the histone proteins, which are the main components of chromatin, the complex of DNA and proteins that make up chromosomes [56]. These modifications (acetylation, phosphorylation, and methylation) play a crucial role in regulating gene expression and chromatin structure, thereby influencing various cellular processes such as DNA replication, repair, and transcription (Figure 5) [57]. These functions can regulate gene expression and play a role in the development of genetic disorders and early development in mammals [58,59,60].

The acetylation of histones is regulated to two enzyme families, histone acetyltransferases (HATs) and histone deacetylases (HDACs), which modify the Lys residues of histones. HATs transfer an acetyl group to the Lys side chain, while HDACs reverse the process [61]. The phosphorylation of histones occurs on serine, threonine, and tyrosine, and is regulated by kinases and phosphatases [62]. The methylation of histones occurs on Lys and arginine side chains, with specific methylases and demethylases [63,64]. Histone modifications regulate multiple biological processes, with one of the most important being early mammalian development. For example, modifications in the 4 and 27 Lys residues of histone 3 (H3K4 and H3K27) regulate the formation of the trophectoderm and the inner cell mass in mice [65], and other histone modifications are related to the regulation of genomic imprinting, *HOX* gene expression, and the regulation of pluripotency in mammalian early development, among others [66].

### 3.3. Small Non-Coding RNA: miRNAs as Epigenetic Regulators

The discovery of the first micro RNA (miRNA), lin-4, in the nematode *Caenorhabditis elegans*, dates back to 1993, but its significance was not fully realized until the discovery of let-7, another miRNA [67,68]. MiRNAs are small RNA molecules, approximately 20–24 nucleotides in length, that do not code for proteins [69,70,71]. MicroRNAs are found in many organisms, including humans, and are involved in various biological processes, such as development, cell proliferation, differentiation, and apoptosis [71,72]. The primary function of miRNAs is to regulate gene expression by binding to the messenger RNA (mRNA) molecules and either degrading them or inhibiting their translation into proteins. The biogenesis of miRNAs begins in the cell nucleus, where they are transcribed from DNA sequences into primary miRNA (pri-miRNA) molecules. These pre-miRNAs are then processed by an enzyme called Drosha to form precursor miRNA (pre-miRNA) hairpin structures. The pre-miRNAs are exported from the nucleus to the cytoplasm by the protein Exportin-5. In the cytoplasm, the pre-miRNAs are further processed by an enzyme called Dicer, resulting in the formation of mature miRNAs. The mature miRNAs are then incorporated into a protein complex called the RNA-induced silencing complex (RISC). Within the RISC, the miRNAs guide the complex to complementary sequences on target mRNA molecules. The binding of miRNAs to mRNA can lead to mRNA degradation or translational repression, depending on the degree of complementarity between the miRNA and the target mRNA (Figure 6) [73]. Other non-coding RNAs were found to be epigenetic regulators. Piwi-interacting RNA (piRNA), small interfering RNA (siRNA), small nucleolar RNA (snoRNA), circular RNAs (circRNAs), and long non-coding RNAs (lncRNAs) can regulate gene expression through various mechanisms, including heterochromatin formation and inhibition of translation [74,75,76]. These RNA molecules were shown to play an important role in shaping the epigenetic landscape [77,78,79]. However, the exact connection between nutrition and the regulation of these non-coding RNAs remains unclear.

## 4. Epigenetic Regulation and Nutrition

Epigenetic regulation plays an important role in milk production in mammals. During lactation, mammary gland cells show extensive epigenetic changes that help to activate the genes required for milk production and secretion. One of the key epigenetic mechanisms involved in milk production is DNA methylation. The methylation of the promoter regions of genes involved in milk synthesis and secretion can lead to the suppression of their expression. The methylation profile regulates milk production [81] and is related to the protein and fat levels in milk. Recently, Wang et al. (2021) demonstrated differentially methylated CpG sites co-located with QTLs for milk protein and fat [82]. The hypomethylation of the activator of transcription (STAT) 5-binding lactation enhancer in bovine lactating mammary glands regulates casein expression [83]. Concretely, an increase in the methylation of the CpG island of STAT5 binding was observed as the post-milking time lengthens and could be related to mammary involution and the decrease in the protein levels in milk [83,84].

Nutrition can affect epigenetic regulation in the mammary gland, which, in turn, can impact milk production. Studies have shown that maternal diet during pregnancy and lactation can influence epigenetic modifications in the mammary gland, leading to changes in the gene expression and milk composition. For example, maternal protein restriction during pregnancy and lactation was shown to alter DNA methylation patterns in the mammary gland of offspring, leading to changes in the expression of genes involved in milk production and secretion in cattle and goats [85,86,87]. The DNA methylation patterns and gene expression in the mammary gland change with the supplementation of certain nutrients, such as choline and folic acid. Some AAs, mainly Met, participate in the one-carbon metabolism that regulates the synthesis of purines and methylation. Other metabolites, such as choline, betaine, and folate, are used to provide methyl donors for methyltransferases, and their use is related to the AA requirements. All of them have an important role in the regulation of S-adenosylmethionine (SAM) [88,89].

The impact of Met supplementation on cows was extensively studied both in vivo and in vitro. Met is an essential AA that plays a crucial role in several biological processes, such as growth and milk production. It is also essential for maintaining the cow’s overall health, as it supports the function of the immune system and acts as an antioxidant, protecting cells from damage caused by free radicals. Enhancing the supply of Met during the peripartum period leads to a greater phosphorylation of the antioxidant transcription regulator (NFE2L2) [90,91,92], along with a reduction in the protein abundance of its negative regulator (KEAP1) in mammary tissue. As a result, the abundance of the antioxidant protein (HMOX1) and various target genes is upregulated. The data suggest the existence of posttranslational and transcriptional mechanisms in mammary tissue during early lactation that likely modulate the antioxidant effect of Met [90]. It was shown that Met supplementation increases the expression of phosphatidylethanolamine methyltransferase (PEMT) and cystathionine β-synthase (CBS), resulting in a higher synthesis of phosphatidylcholine and antioxidants [93]. An enhanced Met supply increases the AKT serine/threonine kinase 1 phosphorylation status and a cascade of intracellular events leading to the upregulation of AAs and glucose transporters [94]. Coleman et al. (2019) studied the effect of increased post ruminal supply of choline during periods of feed restriction-induced negative energy balance (NEB), which was found to have beneficial effects on milk production and liver fatty acid metabolism [95].

The effect of Met supplementation on immunity was studied as well, showing that Met plays a critical role in the immune response, and its deficiency can impair the immune system’s ability to fight infections and diseases [96,97,98]. A strong immune system in cows can help to prevent diseases and infections, leading to an improvement in overall health and productivity. This, in turn, leads to higher milk production, better quality of milk, and lower veterinary costs. In vivo studies provide valuable insights into how Met affects mammary cell function and milk production, informing dietary recommendations and management practices for optimizing cow health and productivity. The supplementation of Met in bovine mammary epithelial cells leads to an increase in milk protein synthesis. This effect is facilitated by the upregulation of gene expression related to several signaling pathways, including cGMP-PKG, Rap1, calcium, cAMP, PI3K-AKT, MAPK, and JAK-STAT [99], and mTOR [100,101]. Qi et al. (2018) demonstrated that Met promotes milk protein and fat synthesis and the proliferation of BMECs through the activation of the SNAT2-PI3K signaling pathway [102].

In addition to DNA methylation, histone modifications also play a role in milk production. The packaging of DNA around histone proteins can be altered by various modifications such as acetylation, methylation, phosphorylation, and ubiquitination, which can affect gene expression. For example, the acetylation of histones is generally associated with active gene transcription, while the methylation of histone can lead to either the activation or repression of genes, depending on the specific modification and location [103]. Histone modifications can also be influenced by nutrition. Studies have shown that dietary components, such as polyphenols and omega-3 fatty acids, can alter histone acetylation patterns in the mammary gland, leading to changes in the gene expression and milk composition in pigs and goats [104,105]. In the context of milk production in cows, it is known that additional carbohydrates can lead to an increased availability of glucose, which can be used by the mammary gland for milk production and milk protein synthesis [106]. However, if the cow’s energy intake from carbohydrates is excessively high, it can result in an imbalance in the cow’s metabolism [107,108,109]. The excess carbohydrates can be converted into fatty acids in the liver through lipogenesis, leading to an increase in fat synthesis [110]. Studies on gene expression related to the amount of energy and fat in the diet suggest that diets with a high energy intake, whether in the form of carbohydrates or fats, could alter the methylation patterns of genes related to fatty acid biosynthesis in the mammary tissue of dairy cows, resulting in a reduction in the milk fat content, as well as the quantity of milk [111,112].

In relation to DNA methylation and histone modifications, non-coding RNAs also regulate gene expression during lactation. MiRNAs can suppress the translation of target genes by binding to complementary sequences in the 3′-untraslated regions (UTR) of mRNAs, leading to the degradation or inhibition of translation. These non-coding RNAs were shown to be influenced by nutrition. For example, maternal diet during pregnancy and lactation can alter miRNA expression patterns in the mammary gland of offspring, leading to changes in the gene expression and milk composition in rats [113,114], and miRNAs are related to DNA methylation regulation. In vitro studies indicate that miRNA-152 is involved in the development of mammary glands and lactation via the regulation of DNMT1 [115], and the inhibition of miRNA-29s provokes the hypermethylation of lactation gene promoters [116]. In relation to milk composition, miRNA-183 is inhibited by prolactin and regulates milk fat metabolism [117], and miRNA-200 is necessary for mammary gland development [118]. Milk composition is also regulated by miRNAs. Cui et al. (2020) found 71 miRNAs differentially expressed in the mammary glands in dairy cows with different milk proteins and fats [119]. At the same, Billa et al. (2021) found eight miRNAs associated with QTLs related to lactalbumin content and fat composition [120]. Finally, other non-coding RNAs were related to mammary gland development and function. For example, Shore et al. (2012) found a non-coding RNA, named pregnancy-induced non-coding RNA (PINC), as a possible regulator of milk production [121]. The high expression of PINC inhibits alveolar cells and prevents milk production and secretion in pregnancy. Recently, Sun et al. (2023) published an extensive review related to circular RNA and its roles in livestock production, including milk production [122]. Table 1 shows the examined nutrients, and the related epigenetic effects and physiological endpoints.

Overall, epigenetic regulation is critical for the proper functioning of the mammary gland during lactation, and the dysregulation of epigenetic mechanisms can contribute to lactation insufficiency or other milk-related disorders. Nutrition can influence epigenetic regulation in the mammary gland, leading to changes in the gene expression and milk production. Understanding the role of nutrition in epigenetic regulation can help to develop strategies for improving milk production and composition, as well as the health and well-being of offspring.

## 5. Conclusions

This review analyzes the diverse effects of protein and AA levels in lactating cow diets on epigenetic regulatory mechanisms. An adequate intake of essential nutrients is necessary for optimal cow health and productivity. Met supplementation was shown to improve milk protein and fat content, milk quality, and milk production, and support the function of the immune system. Additionally, Met plays an important role in reproductive success and overall cow growth and development. Understanding the effects of varying protein and AA levels on epigenetic changes is important, as these interactions impact not only the animal’s productive traits, but also other traits of the animal and its offspring. These epigenetic changes can have significant consequences on the animal’s health, welfare, and productivity. Therefore, it is essential to study the mechanisms underlying the effects of protein and AA levels on epigenetic regulation, and to develop precise and tailored nutritional strategies that optimize animal health, performance, and offspring development.

## Figures and Tables

**Figure 1 animals-13-01883-f001:**
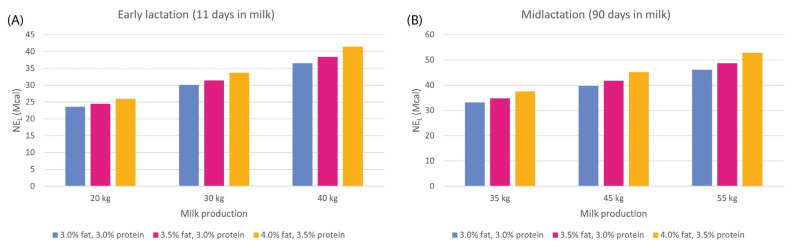
Daily net energy lactation (NE_L_) requirements of large-breed cows (live weight = 680 kg) in early lactation. (**A**) Feed intake estimated on day 11 of its lactation and midlactation. (**B**) Feed intake estimated on day 90 of its lactation in terms of milk production and milk composition [28].

**Figure 2 animals-13-01883-f002:**
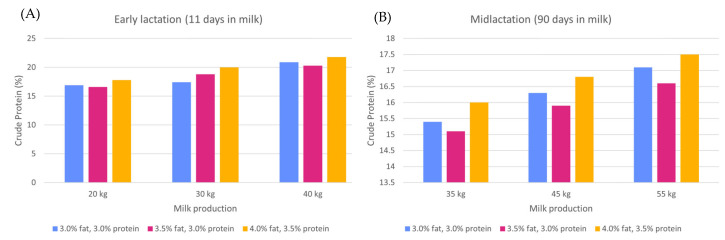
Daily crude protein requirements of large-breed cows (live weight = 680 kg) in early lactation. (**A**) Feed intake estimated on day 11 of its lactation and midlactation. (**B**) Feed intake estimated on day 90 of its lactation in terms of milk production and milk composition [28].

**Figure 3 animals-13-01883-f003:**
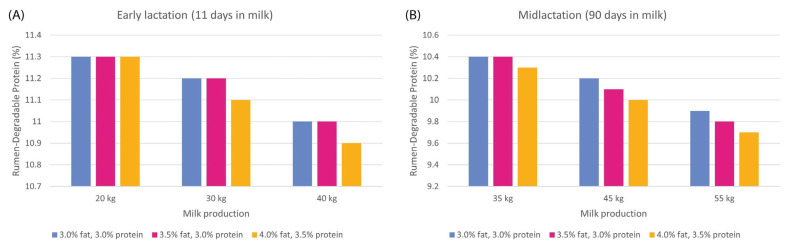
Daily rumen-degradable protein requirements of large-breed cows (live weight = 680 kg) in early lactation. (**A**) Feed intake estimated on day 11 of its lactation and midlactation. (**B**) Feed intake estimated on day 90 of its lactation in terms of milk production and milk composition [28].

**Figure 4 animals-13-01883-f004:**
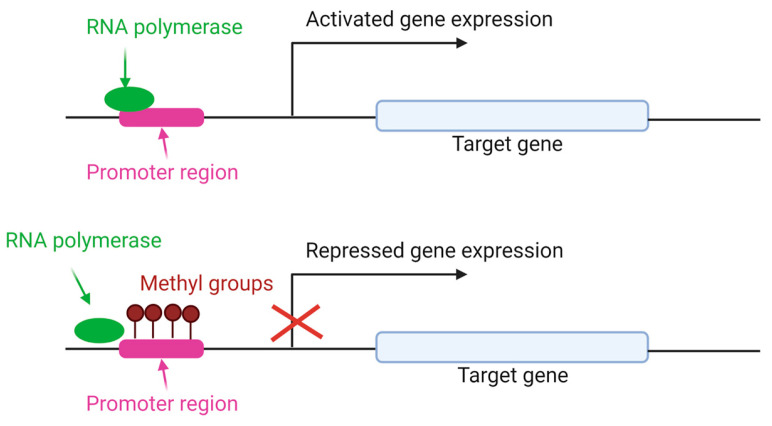
Gene expression regulated by DNA methylation. DNMT enzymes add methyl groups in cytosine residues of CpG island. As a result, RNA polymerase is unable to bind to DNA sequence of gene promoter, and the gene expression is repressed [43].

**Figure 5 animals-13-01883-f005:**
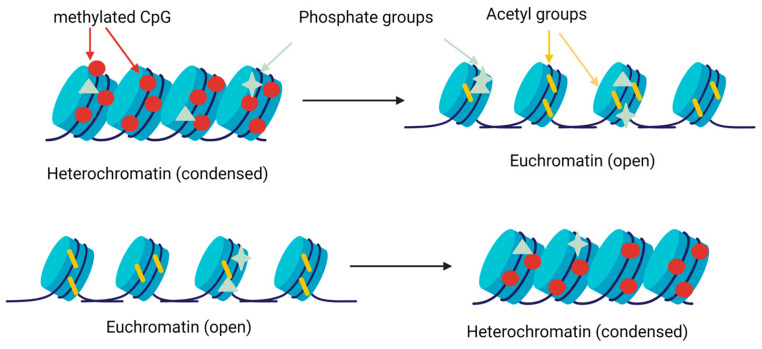
Schematic diagram of the gene expression regulation by histone modifications. Methyltransferases add methyl groups in lysine (Lys) and arginine residues, favoring chromatin condensation and preventing gene expression (up). Acetyltransferases and deacetylases add or remove acetyl groups in Lys residues, decreasing or increasing the condensation of chromatin, respectively (up and down). The kinases and phosphatases add phosphate groups in different locations, changing the condensation of chromatin [43].

**Figure 6 animals-13-01883-f006:**
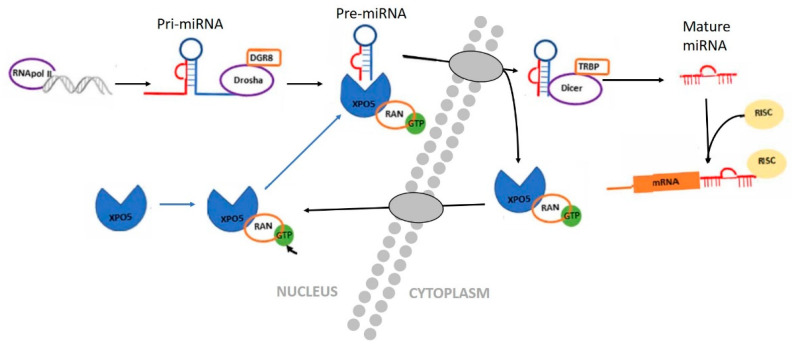
Schematic diagram of miRNA biogenesis. In the nucleus, miRNA is transcribed by RNA polymerase II as primary transcripts (pri-miRNA). The Drosha enzyme cuts this pri-miRNA to form a pre-miRNA, which is actively transported to the cytoplasm by the nuclear transport receptor Exportin-5 (XPO5). In the cytoplasm, the pre-miRNA is cut by a second enzyme, Dicer, to form a mature and short double-stranded miRNA molecule. The miRNA duplex is incorporated into the RISC protein complex [80].

**Table 1 animals-13-01883-t001:** Different epigenetic marks and changes due to changes in diet, and related physiological endpoints.

Nutrient Examined	Epigenetic Mark or Molecule	Physiological Endpoint	References
Protein restriction	DNA methylation patterns altered	Milk production in cattle and goats	[85,86,87]
Methionine supplementation	Phosphorylation of NFE2L2	Antioxidant effect in cattle	[90,91,92]
	Upregulation of gene expression	Protein and fat synthesis in milk in cattle	[102,103,104]
Choline supplementation	DNA methylation patterns altered	Milk production and liver fatty acid metabolism in cattle	[97]
Polyphenol and omega-3 fatty acid supplementation	Histone acetylation patterns	Milk composition in pigs and goats	[104,105]
Carbohydrates and fat supplementation	DNA methylation patterns altered	Reduction in milk fat content; milk production in cattle	[111,112]
High fat supplementation	MiRNA expression patterns altered	Milk production in rats	[115]
Energy restriction	MiRNA expression patterns altered	Milk production and composition in cattle	[120]

## Data Availability

Data sharing not applicable.

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
