# Peer review of "How Does Nutrition Affect the Epigenetic Changes in Dairy Cows?"

_animals, 2023, doi:10.3390/ani13111883_

Round 1
Reviewer 1 Report
The subject of animals-2412087 is Epigenetic alterations due to nutrition in dairy cows. However, only L243-262 are directly related to epigenetic changes in dairy cows. L66-139 are dairy cow nutrition and are not associated with epigenetic regulators. L140-241 are descriptions of common epigenetic regulators. L262-342 is not directly related to epigenetic changes in dairy cows. Therefore, animals-2412087 needs restructuring.
Author Response
The subject of animals-2412087 is Epigenetic alterations due to nutrition in dairy cows. However, only L243-262 are directly related to epigenetic changes in dairy cows. L66-139 are dairy cow nutrition and are not associated with epigenetic regulators. L140-241 are descriptions of common epigenetic regulators. L262-342 is not directly related to epigenetic changes in dairy cows.
Therefore, animals-2412087 needs restructuring.
Dear reviewer,
Thank you very much for your appreciated comments. We fully agree with your comments. With this review we wanted to analyse the effects that nutrition could have on dairy cows at the level of possible epigenetic changes, so we have considered it important, first, to provide information about this nutrition and epigenetic mechanisms in general, with the aim of so that any researcher,
even if not a specialist in the area of study, could clearly understand the work.
Perhaps the title of the paper was not the most appropriate, so we have thought to title the paper as "How does nutrition affect the epigenetic changes in dairy cows?", to facilitate the reviewer's understanding of the topic of the paper.
Reviewer 2 Report
The author provides evidence connecting daily diet nutrition to the molecular mechanism of epigenetic regulation. Studying how the diet affects epigenetic regulation can make it more straightforward for people, especially industry and farm people to understand how to improve the dairy cow growing conditions. Most conclusions cited in the paper are recently studied, so I think this review can give an advanced guideline for industry or farm.
Author Response
The author provides evidence connecting daily diet nutrition to the molecular mechanism of epigenetic regulation. Studying how the diet affects epigenetic regulation can make it more straightforward for people, especially industry and farm people to understand how to improve the dairy cow growing conditions. Most conclusions cited in the paper are recently studied, so I think
this review can give an advanced guideline for industry or farm.
Dear reviewer,
Thank you very much for your comments.
Reviewer 3 Report
The manuscript is a general overview of epigenetics and dairy cow nutrition. It is easy to ready but too general to help researchers in the field. When a topic is introduced a few examples are then given, but it would be more helpful to have a summary table of some of the key findings in the field without reproducing previous reviews. The table should have information about the nutrients examined, the epigenetic mark or molecule involved and the physiological endpoint evaluated.
Figures 4 and 5 are common knowledge and do not add to the review. I think these should be replaced with a summary figure of the impact of nutrition and epigenetic alterations on
Author Response
The manuscript is a general overview of epigenetics and dairy cow nutrition. It is easy to ready but too general to help researchers in the field. When a topic is introduced, a few examples are then given, but it would be more helpful to have a summary table of some of the key findings in the field without reproducing previous reviews. The table should have information about the nutrients examined, the epigenetic mark or molecule involved, and the physiological endpoint evaluated.
Figures 4 and 5 are common knowledge and do not add to the review. I think these should be replaced with a summary figure of the impact of nutrition and epigenetic alterations on.
Dear reviewer,
Thank you very much for your comments and suggestions. Although figures 4 and 5 could be considered common knowledge, we believe that they can help researchers who are not experts in the field, to understand the epigenetic mechanisms that could be involved.
On the other hand, following your indications, we have added a a table summarizing the nutrients examined, the epigenetic change involved, and the physiological end point evaluated (table 1).
Reviewer 4 Report
Authors in submitted MS linked epigenetic alteration and nutrition in dairy caws. The MS is systematic and very interesting.
My minor suggestion is:
Could you please increase resolution on Figs 1-3, and is it possible to show statistic on those graphs?
Author Response
Authors in submitted MS linked epigenetic alteration and nutrition in dairy caws. The MS is systematic and very interesting.
My minor suggestion is:
Could you please increase resolution on Figs 1-3, and is it possible to show statistic on those graphs?
Dear reviewer,
Thank you very much for recommendation. Following that, we have improved the quality of the figures 1-3
Reviewer 5 Report
This is an interesting article regarding nutrition and its effect on the DNA of cattle. It contains some interesting information, but does not follow the suggested guidelines for a review- which includes the use of the PRISMA statement which is not in the manuscript. How were manuscripts chosen for inclusion within the report are unknown.
In addition, there are multiple grammatical issues within the manuscript, and some of the figures are unclear.
Further comments are below
Line 17- isn’t this dairy cattle rather than beef cattle?
Line 30- again, isn’t this dairy cattle rather than beef cattle?
Line 36- Cattle produce 81% of the worlds milk, and this is estimated to be at around 930 …. (reword)
Line 38- an increase in world milk … (reword)
Line 51- also other trains such as health … (reword)
Line 54- and improve cattle health …. (reword)
Line 57- mechanisms regulate gene expression independently of the DNA sequence …. (reword)
Line 59- high levels of protein in a rat diet leads to an increase in mRNA expression …. (reword)
Line 60- humans, methyl group donors such as betaine could …(reword)
Line 64- during the lactating period …(reword)
Line 70- required by lactating dairy cows …(reword)
Figure 1-3- this is unclear and may benefit from some further explanation
Line 86- and to regain its lost weight … (Reword)
Line 121- AA requirements in dairy cattle …(reword)
Line 127- To meet the AA requirements, … (reword)
Line 133- if AA levels are balanced …. (reword)
Figure 3- there is a bit missing from figure 3A
Line 151- identified in both humans and mice
Line 172- methyl group donors such as Met … (reword)
Line 179- related to the presence …. (reword)
Line 213- in vivo needs to be in italics
Line 244- mammary glad cells undergo extensive … (reword)
Line 248- milk production in cows …. (reword)
Line 251- this doesn’t make sense- please reword as I cannot suggest an alternative as I don’t know what you want to include
Line 253- of CpG (delete in)
Line 265- in the one carbon unit … (reword)
Line 267- their use is related to the AA requirements …. (reword)
Line 270- essential AA that plays …(Reword)
Line 280- PEMT and CBS need defining
Line 280- CBS resulting in a higher …(Reword)
Line 305- not sure what you mean by reexpression of gene expression- please reword
Line 321- Related to DNA methylation …(Reword)
Line 323- untranslated is a typo
Line 332- necessary for mammary gland development …(reword)
Line 334- expressed in the mammary gland ..(Reword)
Line 362- and AA levels on epigenetic …(reword)
There are some suggestions detailed above
Author Response
Dear reviewer,
Thank you very much for your suggestions and comments. The answers are in the attach,

Round 2
Reviewer 1 Report
I understood that the authors emphasized not only the main objectives, but also a review of dairy cow nutrition (particularly proteins and amino acids) and a review of epigenetic regulators. The revised title is more representative of the manuscript content than the previous one. The addition of Table 1 makes it easier to understand the relationship between dairy cow nutrition and epigenetic marks. I thought the manuscript was well improved.
However, please consider the following:
1) The addition of information on the animals tested in the experiments cited in Table 1 and L248-351 would enhance the value of this paper.
2) Aren't [102,103,104] in Table 1 and L303-308 not epigenetic regulation but normal gene expression changes?
Author Response
Dear reviewer,
Thank you very much for your comments. We answer below:
I understood that the authors emphasized not only the main objectives, but also a review of
dairy cow nutrition (particularly proteins and amino acids) and a review of epigenetic regulators.
The revised title is more representative of the manuscript content than the previous one. The
addition of Table 1 makes it easier to understand the relationship between dairy cow nutrition
and epigenetic marks. I thought the manuscript was well improved.
However, please consider the following:
1) The addition of information on the animals tested in the experiments cited in Table 1 and L248-
351 would enhance the value of this paper.
The species related to this information have been added.
2) Aren't [102,103,104] in Table 1 and L303-308 not epigenetic regulation but normal gene
expression changes?
In these cases, the expression genes changes are demonstrated. However, these expression
genes changes are related to mTOR signaling pathways, which is regulated by epigenetic
mechanisms (DOI: 10.1016/j.jmb.2018.10.008).
Reviewer 5 Report
I wish to thank the authors for addressing the comments diligently, and wish them the best of luck for their future studies
Author Response
Dear reviewer,
Thank you very much for your comments and suggestions.
Regards,